# Comprehensive Characterization of the Coding and Non-Coding Single Nucleotide Polymorphisms in the Tumor Protein p63 (TP63) Gene Using In Silico Tools

**DOI:** 10.3390/biom11111733

**Published:** 2021-11-20

**Authors:** Shamima Akter, Shafaat Hossain, Md. Ackas Ali, Md. Ismail Hosen, Hossain Uddin Shekhar

**Affiliations:** 1Department of Bioinformatics and Computational Biology, George Mason University, Fairfax, VA 22030, USA; sakter5@gmu.edu; 2Clinical Biochemistry and Translational Medicine Laboratory, Department of Biochemistry and Molecular Biology, University of Dhaka, Dhaka 1000, Bangladesh; shossain16@uh.edu (S.H.); ismail.hosen@du.ac.bd (M.I.H.); 3Division of Computer Aided Drug-Design, The Red-Green Research Center, 16, Tejkunipara, Tejgaon, Dhaka 1215, Bangladesh; akashali16dec@gmail.com

**Keywords:** TP63, nsSNPs, non-coding SNPs, dbSNP, MD simulations, molecular docking, I-Mutant 2.0, pathogenic prediction, computational biology, bioinformatics

## Abstract

Single nucleotide polymorphisms (SNPs) help to understand the phenotypic variations in humans. Genome-wide association studies (GWAS) have identified SNPs located in the tumor protein 63 (TP63) locus to be associated with the genetic susceptibility of cancers. However, there is a lack of in-depth characterization of the structural and functional impacts of the SNPs located at the *TP63* gene. The current study was designed for the comprehensive characterization of the coding and non-coding SNPs in the human *TP63* gene for their functional and structural significance. The functional and structural effects of the SNPs were investigated using a wide variety of computational tools and approaches, including molecular dynamics (MD) simulation. The deleterious impact of eight nonsynonymous SNPs (nsSNPs) affecting protein stability, structure, and functions was measured by using 13 bioinformatics tools. These eight nsSNPs are in highly conserved positions in protein and were predicted to decrease protein stability and have a deleterious impact on the TP63 protein function. Molecular docking analysis showed five nsSNPs to reduce the binding affinity of TP63 protein to DNA with significant results for three SNPs (R319H, G349E, and C347F). Further, MD simulations revealed the possible disruption of TP63 and DNA binding, hampering the essential protein function. PolymiRTS study found five non-coding SNPs in miRNA binding sites, and the GTEx portal recognized five eQTLs SNPs in single tissue of the lung, heart (LV), and cerebral hemisphere (brain). Characterized nsSNPs and non-coding SNPs will help researchers to focus on *TP63* gene loci and ascertain their association with certain diseases.

## 1. Introduction

Human tumor protein 63 is encoded by the *TP63* gene, which is 4944 bp (4.94 kb) long and located in chromosome 3 at the 3q28 locus. It has 12 isoforms listed in the UniProt database. TP63 isoform 1 is 680 amino acids long and acts as a transcription factor (TF) that regulates gene expressions in multiple pathways, notably in tumorigenesis and development [1]. TP63 plays an essential role in the development of the body’s organs and tissues. Aside from its developmental roles, the p63 protein appears to be required for the preservation of various cells and tissues in late age. The TP63 protein can function as a transcriptional activator or repressor and binds to DNA in a sequence-specific manner [2]. TP63 shares sequence similarity with the tumor suppressor p53 family, with cellular homeostasis, development, differentiation, and growth control as the crucial functions [3,4].

Several genome-wide association studies (GWAS) and case-control studies have shown the association of single nucleotide polymorphisms (SNPs) in the human TP63 locus with various cancers, i.e., lung cancer, head and neck cancer, colon cancer, urinary bladder cancer, and other diseases [5,6,7,8,9,10]. The difference of a single nucleotide, i.e., A, T, C, or G at a specific position in the genome is defined as an SNP. Human genes contain about 93% SNPs [11]. Coding, non-coding, or intergenic regions of genes have SNPs [12]. In the human genome, SNPs are the most common type of genetic variation occurring both in the coding and non-coding regions of a gene. Nonsynonymous SNPs (nsSNPs) occur within a coding region and have pathogenic impacts on the protein structure, function, stability, and solubility through amino acid replacement in the protein sequence [11,13]. These nsSNPs can be categorized as either pathogenic/deleterious, causing disease phenotypes, or tolerated/neutral, causing no effect on protein structure and function [14,15]. Non-coding SNPs (5′ UTR and 3′ UTR SNPs and intron SNPs) play important roles in gene expression, variation, and gene regulation that affect the translation of the protein. Subsequently, this outcome leads to altered protein production. Hence, non-coding SNPs of the *TP63* gene can assist in identifying the altered gene expression together with gene regulation.

Different computational tools and approaches to predict deleterious SNPs and their roles in protein stability, function, and structure are extensively used [16,17,18,19]. One study investigated the nsSNPs of the human *CHK2* gene using computational tools to predict the structural and functional impacts on the protein [20]. In another study, researchers applied several computational tools along with molecular docking and MD simulations to predict the structural and functional impacts of SNPs in the human *STK11* gene [21]. Similarly, nsSNPs in the human *hTERT* gene were analyzed for their structural and functional significance through in silico tools [22]. Overall, these studies revealed the effectiveness and usage of computational tools to characterize the genetic variants in a target gene of interest. Several other studies also implemented in silico tools differently along with molecular docking and MD simulations to characterize nsSNPs in different genes [23,24] and to design drugs for COVID-19 [25,26]. However, only limited studies have explored both nsSNPs and non-coding SNPs in different genes, and few studies have implemented molecular docking and molecular dynamics (MD) simulation approaches to validate nsSNPs. Moreover, no study has been performed for the analysis of nsSNPs with molecular docking and MD simulation approaches for the TP63 protein. Therefore, the main goals of this study were as follows:Determine the consequences of various nsSNPs in the human *TP63* gene on the TP63 protein using different in silico tools.Evaluate the effect of nsSNPs on the binding affinity of the TP63 protein with its ligands (DNA) by molecular docking to confirm the consequences.Simulate interactions of DNA and TP63 protein with molecular dynamics simulations to validate the effect on protein function caused by high impact nsSNPs.

Furthermore, the study also aimed to analyze non-coding SNPs in the *TP63* gene through RegulomeDB, PolymiRTS, and GTEx portal to investigate the functionally important non-coding SNPs in this locus.

## 2. Methods and Materials

We retrieved the SNP data from the ENSEMBLE genome database and obtained 64,144 SNPs in both the coding and non-coding categories. Then, we filtered missense SNPs, which are described as nsSNPs of the coding region in the TP63 protein. Intron, 5′ UTR, and 3′ UTR SNPs were also selected for non-coding SNPs (836) of the TP63 protein. The functional consequence analysis was carried out using nine different in silico tools. First, 28 nsSNPs were retrieved through analysis with SIFT, Polyphen2, and CADD. Later, 17 nsSNPs were obtained after analysis with six in silico tools (i.e., PROVEAN, ClinVar, MutPred2, PANTHER, PhD-SNP, and SNPs&GO). The structural impact analysis was accomplished by implementing ConSurf, HOPE, I-Mutant, and Mutation 3D, and nine nsSNPs were found with significant deleterious effects. Further, molecular docking analysis was performed. Docking analyses were chosen to perform MD simulations to verify the outcome of the predictions. In the case of non-coding SNPs, Regulome DB analysis and GTEx portal were utilized along with PolymiRTS analysis. A flowchart showing the different steps of this study is presented in Figure 1.

### 2.1. SNP Data Retrieval

In this study, the human *TP63* gene was investigated in the ENSEMBL genome browser [27]. The transcript encoding the full-length TP63 protein (680 amino acids) was selected to retrieve SNP data [28]. Further, SNPs were downloaded from the dbSNP database [29]. 

### 2.2. Sequence Investigation (Functional Consequence Analysis of nsSNPs)

Sorting Intolerant From Tolerant (SIFT) was used to detect the deleterious nsSNPs. SIFT can differentiate functionally neutral amino acid changes from functionally deleterious ones [30]. This software presumes that major amino acids will be retained; shifts at a particular position appear to be predicted as deleterious in proteins. If the normalized probability is less than the tolerance value (0.05), then substitutions are considered as “deleterious” and equal or higher than 0.05 is considered as “tolerated”. Reference SNP IDs (rsIDs) of each variant of the human TP63 protein (UniProt ID: Q9H3D4) and individual amino acid substitutions were provided as input in the SIFT tool, and the resulting score values along with their interpretations were recorded.

Polymorphism Phenotyping v2 (PolyPhen2) uses Naive Bayes to classify and predict the functional impacts of allele modifications [31]. For each variant, PolyPhen2 estimates the position-specific independent count (PSIC) based on site-specific sequence conservation along with the difference of scores between the native and mutant variants. PolyPhen2 classifies the SNPs into 3 different classes: (1) benign, (2) possibly damaging, or (3) probably damaging. The input on the PolyPhen2 web server includes the FASTA sequence of human TP63 protein and the individual amino acid substitutions.

Combined Annotation-Dependent Depletion (CADD) [32] is an SNP predicting algorithm that prioritizes causal variants in polymorphism studies. CADD integrates multiple annotations in a single metric by comparing variants that have survived in a natural selection to simulate mutations. CADD prioritizes causal variations in both research and clinical settings. Chromosomal locations of the human TP63 variants are given as input to the CADD web server.

Protein Variation Effect Analyzer (PROVEAN) is a server that assesses the functional impact of a substituted amino acid or insertion-deletion mutation on a protein across organisms. PROVEAN can provide high-throughput analysis at both the genomic and protein levels for human and mouse variants [33]. For each variant, PROVEAN calculates a score from the alignment of homologous sequences, and a score of −2.5 is considered damaging. Input in the PROVEAN web server was the FASTA sequence of human TP63 protein and changes in single amino acids (nsSNPs). The TP63 protein sequence from the NCBI database was the input sequence for PROVEAN. A cutoff score of −2.5 was selected for deleteriousness.

SNPs&GO uses functional annotations of proteins to assess the impact of single amino acid substitutions [34]. SNPs&GO analysis provides the prediction of SNPs using three tools, i.e., SNP&GO, PhD-SNP, and PANTEHR. SNPs&GO utilizes support vector machines, and input includes the sequence or its three-dimensional protein structure, target variations, and gene ontology (GO) term functionality. The output of the algorithm provides the probabilities of association of each SNP with disease(s).

ClinVar helps to analyze human genetic variations along with annotation of variant disease associations. ClinVar aggregates the known variant interpretations and makes them publicly available on the ClinVar database [35]. The ClinVar database was searched by individual amino acid changes to identify disease-associated variants.

MutPred2 enhances pathogenic amino acid substitution prioritization, predicts potentially disease-causing molecular pathways, and returns interpretable distributions of the pathogenicity score on individual genomes [36]. Input in the MutPred2 web server was the FASTA sequence of human TP63 protein and changes in single amino acids (nsSNPs).

InterPro and NCBI domain prediction tools employ protein families to predict domains of protein by evaluating them functionally. The Conserved Domain Search tools in NCBI [37] and InterPro [38] were used to identify the domains of the TP63 protein. nsSNPs were positioned in three domains of the protein. For both domain searching tools, the input query was the FASTA amino acid sequence of the TP63 protein. The NCBI domain search tool used Pfam superfamily classification and InterPro used different classification programs such as Pfam, InterPro, and others to identify the domains and regions in TP63.

I-Mutant 2.0 is an algorithm that uses a support vector machine to provide an assessment of protein stability change for the variations in single nucleotides [39]. I-Mutant 2.0 uses either protein structure or protein sequence for predictions and estimates the shift in protein stability, and simultaneously predicts the corresponding Delta G (∆∆G) values. ∆∆G or double changes intended is a measurement for determining the effects of a single point mutation on protein stability. It is considered to be a good predictor of whether or not a point mutation will improve the stability of the protein. The sequence of the human TP63 protein and individual amino acid substitutions were given as input in the I-Mutant 2.0 web server.

As an algorithm and web server, Mutation3D detects changes in amino acids within protein three-dimensional structures. Mutation3D can distinguish functional and nonfunctional variations [40]. The mutation3D web server enables users to examine the substitution in several common formats while offering easy access to examine mutation clusters derived from 6811 cancer sequencing studies reported from over 975,000 somatic mutations.

ConSurf web server was utilized to recognize evolutionarily conserved amino acid residues (native) and to classify nsSNPs in each position [41]. According to the evolutionary relationship, the ConSurf server determines the evolutionary conservation rate of the amino acid positions in a protein molecule through a user-submitted protein sequence and its homologs. Consurf is a tool that gathers information to perform phylogenetic analysis using the empirical Bayesian method to calculate the conservation scores, which are divided into nine discrete bins. Bin 9 refers to the most conserved positions and bin 1 indicates the least conserved positions. More exclusively, conservation scores ranging from 7–9, 5–6, and 1–4 denote the high, middle, and low conserved amino acids, respectively.

HOPE analyzes functional and structural impacts of point mutations. HOPE incorporates data from multiple information sources including measurements of 3D protein coordinates using services of WHAT IF Web, sequence annotations at UniProt database, and predictions of DAS service [42]. Data stored with these sources are used to classify the effects of a mutation on both protein function and three-dimensional structure using a decision scheme. With the help of text, statistics, and animations, HOPE produces a report that is simple to use and easily understandable. 

### 2.3. Structural Modeling

Phyre2 [43] was used to predict the structure of TP63. Phyre2 predicted the 2RMN (PDB ID:2rmn) structure, which is a DNA binding domain of TP63. 2RMN was also obtained in PDBSum server [44] by searching with the full sequence of TP63 and generating the full PROCHECK analyses with the RAMACHANDRAN plot (Appendix A). PROCHECK statistics and plots have been supplied in the Appendix A. This 3D structure was extensively used to generate wild-type and mutant peptide structures of nsSNPs for molecular docking analysis.

### 2.4. Molecular Docking

Molecular docking was performed to assess the impacts of nsSNPs on the interactions between DNA and TP63. The DNA is a direct ligand of the TP63 protein. Binding affinity is a measurement of the interactions between protein molecules and ligands, protein, peptide, or DNA. In this study, molecular docking was performed on the AutoDock Vina platform using virtual screening tools -PyRx [45]. 3QYN (i.e., PDB ID:3qyn, which is a complex DNA binding domain of TP63 and two DNA chains) was taken for the analysis. DNA structure was extracted, and a single chain was chosen using USCF Chimera 1.14 to perform docking analysis. In DNA–protein docking, the DNA chain was used as a macromolecule. Peptides of wild-type TP63 protein and mutant TP63 protein were used as ligands. A similar approach was followed in another study where deleterious effects of nsSNPs of the human *RASSF5* gene on protein structure and function were predicted by employing in silico analysis [46]. All wild-type and mutant peptides of TP63 were generated using the build structure tool in Chimera 1.14 [47] with energy minimization. The DNA molecule and ligands were uploaded in PyRx and subsequently converted into PDBQT (.pdbqt) format using Autodock Vina [48]. We also put the grid box parameters as follows: X = 41.2, Y = 52.0, Z = 50.0. The binding interactions were visualized in Chimera 1.14.

### 2.5. Molecular Dynamics Simulation

Molecular dynamics (MD) simulation was performed to observe the binding interactions of wild-type and mutant TP63 protein with DNA to obtain insight into the structural dynamics and stability for 250 ns. The YASARA Dynamics suite [49] was used for performing MD simulation, and the AMBER14 force field [50] was employed to describe the macromolecular system. In the beginning, the TP63 peptide–DNA complexes were cleaned together with the optimization of the H-bond network. Then, the grid size of 96.96 × 96.96 × 96.96 Å was set in a cubic box with the conditions of the periodic boundary. The ionic strength of Na^+^/Cl^−^ was 0.9% for neutralizing the system at 310 K and pH 7.4. The temperature was simulated using the Berendsen thermostat where the pressure was kept constant throughout the process. A periodic boundary condition was incorporated to perform the simulation. The particle-mesh Ewald method [51] was used for long-range electrostatic interaction calculations at a cut-off distance of 8 Å. Multiple time-step algorithms were employed where the simulation time step was selected as 1.25 fs [52]. Finally, MD simulation was performed for 250 ns, and snapshots were saved at every 100 ps into MD trajectory for analysis. Bond distance, bond angle, dihedral angles, coulombic and van der Waals interactions, solvent-accessible surface area (SASA), radius of gyration (Rg), and root-mean-square-deviation (RMSD) of the complexes were collected from the MD simulations.

### 2.6. Analysis of the Functional Consequences of Non-Coding SNPs

RegulomeDB offers an annotation of regulatory SNPs and integrates the information from experimental data sets, computational predictions, and manual annotations from ENCODE. This tool assigns scores to variants to distinguish the functional SNPs from a wide pool [53]. rsIDs of the individual variants were submitted to the RegulomeDB database to assess the consequences of noncoding SNPs.

Polymorphism in microRNAs and their Target Sites (PolymiRTS) database 3.0 incorporates data from ligation, hybrids sequence, and crosslink experiments to analyze miRNA–mRNA interactions [54]. The PolymiRTS database analyzes the functional consequences of SNPs in miRNA target sites and seed regions. rsIDs of the individual variants were submitted to the PolymiRTS database to assess the consequences of noncoding SNPs.

Genotype-Tissue Expression (GTEx) finds the association between genetic alterations and gene expression in human tissues. GTEx relates these regulatory mechanisms to traits and diseases [55]. rsIDs of the individual variants were submitted to the GTEx database to assess the consequences of noncoding SNPs.

## 3. Results

### 3.1. TP63 SNP Data Retrieval

The SNP data for the *TP63* gene contained a total of 64,144 SNPs with transcript ID: ENST00000264731.8. Among all, different SNPs were as follows: (1) 455 missense or nonsynonymous SNPs (nsSNPs), (2) 242 synonymous SNPs, (3) 680 UTR SNPs, (4) 62,727 intron SNPs, (5) 11 frameshift SNPs, (6) 6 inframe SNPs, (7) 11 stop-gained SNPs, and (8) 12 other SNPs. Different categories of SNPs are presented in Appendix A.

### 3.2. Prediction of Functionally Important nsSNPs in the TP63 Gene

A variety of tools were selected for computational analysis of the nsSNPs of TP63. Initially, SIFT, PolyPhen2, and CADD were used for the analysis. Out of 455 nsSNPs, 194 SNPs were predicted to be deleterious by SIFT. Probably damaging criteria were selected using PolyPhen2, and 121 nsSNPs were found as deleterious. Later, CADD was used to predict the nsSNPs. Likely deleterious criteria were chosen, and 28 nsSNPs were predicted to be deleterious (Appendix A). Further, 28 nsSNPs were selected for analysis using PROVEAN, CLinVar, SNPs&GO, and MutPred2. A total of 23 SNPs were predicted to be deleterious/pathogenic, and the remaining five SNPs were detected as neutral by PROVEAN. ClinVar evaluated nine nsSNPs as pathogenic, three nsSNPs as likely pathogenic, three nsSNPs with uncertain significance. The rest of the nsSNPs were not found in the ClinVar repository (Appendix A).

SNPs&GO simultaneously performed the analysis using three different tools: PhD-SNP, PANTHER, and SNPs&GO, and provided results separately along with probability scores (Table 1). The number of deleterious nsSNPs predicted by SNPs&GO, PhD-SNP, and PANTHER was 28, 23, and 24, respectively (Figure 2A). MutPred2 predicted the structural/functional effect such as gain or loss of a definite structure or function in domains of TP63 due to SNPs. Through this analysis, seven nsSNPs were found to cause no gain or loss of structure in protein, describing no effect due to SNPs. The remaining 21 nsSNPs were shown to cause gain or loss of structural parts of a specific protein.

Different types of gain/loss of structure and function/altered function for protein TP63 were observed after analysis with MutPred2. The impacts in MutPred2, such as loss of strand, gain of strand/intrinsic disorder, altered ordered interface, altered DNA binding, altered stability, altered metal binding, etc., were found with significant p-value and high MutPred2 scores (Table 1). MutPred2 evaluated nine DNA binding domain SNPs with high probability scores and altered structure and functions.

### 3.3. Domain Identification for nsSNPs

The NCBI conserved domain search tool provided three domains of TP63 protein as follows; (1) P53 DNA binding domain (177–358 amino acid residues) usually binds to the DNA, (2) P53_tetramer domain (392–428 amino acid residues), which is described as P53 tetramerization motif according to the Pfam protein family, and (3) SAM_tumor_63 (543–607 amino acid residues), which is defined as SAM domain of tumor-p63 proteins, where SAM stands for sterile alpha motif (Appendix A). InterPro defined these three domains as p53_DNA_bd (DNA binding), p53_tetrameristn, and SAM domain, respectively. Besides these domains, UniProt provided four regions: (1) transcription activation, (2) interaction with HIPK21, (3) oligomerization, and (4) transactivation inhibition. These domains and regions have specific lengths or regions in the protein and are summarized in Figure 3.

SIFT, PolyPhen-2, CADD, PROVEAN, ClinVar, MutPred2, SNPs&GO, PhD-SNP, and PANTHER were evaluated as pathogenic/deleterious 17 nsSNPs that are positioned in three principal domains (Figure 3). Among them, eight nsSNPs were predicted as disease-causing by all in silico tools (Figure 2B). Prioritizing the nsSNPs within the domains of TP63, these 17 nsSNPs were selected for structural analysis with I-Mutant 2.0, ConSurf, Mutation 3D, and HOPE.

### 3.4. Structural Analysis

#### 3.4.1. I-Mutant 2.0 Analysis

The selected 17 nsSNPs of the TP63 protein were then chosen to check for protein stability through analysis with I-Mutant 2.0. All 17 nsSNPs were predicted to decrease the stability of the TP63 protein by I-Mutant 2.0 (Table 2), as DDG/∆∆G was negative for all nsSNPs run by this tool.

#### 3.4.2. Effect of nsSNPs on Evolutionary Conservation of TP63 Protein Using Consurf

Seventeen nsSNPs were submitted for ConSurf analysis. 11 wild-type residues (R266, R319, R337, R343, R376, R408, G349, D351, D355, R647, and R655) were observed as highly conserved. Among them, R319 is a structural and buried residue, and the others are functional and exposed (Figure 4A). The structural residue plays an important role in the protein conformation or the folding of TP63. The R319 residue is buried because of the protein’s hydrophobic core. Three residues were found to be medium conserved; two residues (R318, R379) are functional and exposed, and one (C347) is buried. The remaining residue (L562R) is not conserved (Figure 4B). ConSurf analysis showed that nsSNPs R408C, R408H, C347F, D351G, D355N, G439E, R266Q, R318H, R319H, R337Q, R343Q, R379C, R379C, L562R, R647H, and 655Q are in highly conserved residues are deleterious for the TP63 protein’s structure and function.

#### 3.4.3. Mutation 3D Analysis

Seventeen nsSNPs were selected to perform 3D analysis using the Mutation 3D tool. This analysis showed the three-dimensional structure of 153–388 residue length (available in PDB with 100% sequence similarity) TP63 protein with 12 nsSNPs. Nine SNPs are present in the DNA binding domain of TP63. The other SNPs are in two different domains: one is the p53 tetramer domain, and the other is the SAM-2 domain (Appendix A).

#### 3.4.4. HOPE Analysis

Seventeen SNPs were chosen for evaluation using the HOPE web server tool. For each nsSNP, there was a change in size, charge, and hydrophobicity of the amino acid residues in the respective positions of the TP63 protein. The size of amino acid residues was changed to large/small and the charge was observed to lose/gain. Increased/decreased hydrophobicity was also observed in the protein through the loss of hydrogen bonds due to nsSNPs. These changes disrupt the correct folding of the protein. Results from HOPE analysis are presented in Table 3. Hence, (1) regulation of the protein’s activity through the signal transfer from binding domain to activity domain and (2) binding of other molecules, are hampered. These effects impede the protein’s function overall.

C347F showed loss of di-sulfide bridge that affects the stability of the TP63 protein structure. R343Q, R266Q, R319H, R337Q, and R318H were observed to cause (1) smaller sizes of amino acid residue, (2) loss of positive charge, and (3) increased hydrophobicity in the protein. Glycine (G) is very flexible and possesses a particular conformation for the TP63 protein; D351G or G349E disrupts the protein structure. HOPE created images for the nine SNPs using a partial 3D structure of the TP63 protein (PDB ID: 3QYN). These SNPs are present in the DNA binding domain showing the wild-type and mutated residues at one position with two different colors (Figure 5). Therefore, we decided to perform molecular docking for these nine DNA binding domain SNPs.

### 3.5. Structural Effect Analysis of nsSNPs

#### Molecular Docking

The DNA binding domain structure of TP63 protein is an available PDB structure, and the DNA is a direct ligand for the TP63 protein. Thus, DNA binding domain nsSNPs were selected for molecular docking analysis. Furthermore, these nsSNPs were evaluated as pathogenic by all in silico tools, and HOPE created the image for these SNPs. 

The results of molecular docking showed the binding affinity of mutated and native TP63 peptides (ligands) towards DNA and determined the consequences of nsSNPs on TP63–DNA interactions. Five nsSNPs (R266Q, R319H, C347F, G349E, and D351G) were shown to decrease in binding affinity compared to wild-type peptides while interacting with DNA (Table 4). The interaction patterns of the docked DNA–protein complexes were visualized and studied in USCF Chimera 1.14 (Figure 6A–D). DNA–TP63 complexes are shown in Figure 7, where peptides are in the active site of the DNA. G349E showed a decrease in binding affinity with DNA by losing H-bonds. G349 and R319 formed seven and eight bonds, respectively, whereas E349 and H319 created three and four bonds, respectively, with the DNA. Wild-type peptide (G349) showed a binding affinity of −6.4 kcal/mol, which was reduced to −5.8 KJ/mol due to mutation. Thus, a significant reduction was observed for R319H, C347F, and G349E. Finally, these three nsSNPs were selected for MD simulations.

### 3.6. Molecular Dynamics (MD) Simulations

A 250 ns molecular dynamics simulation was conducted to study the deviation of binding interactions of mutant TP63 proteins (E349, F347, H319) with DNA from native TP63 proteins (C347, R319, and G349) with DNA. MD simulations were performed in terms of root mean square deviation (RMSD), Rg, SASA, and H-bond analysis to observe the deviations of structural dynamics between native and mutant(s)in normal physiological conditions. RMSD values were estimated for native and mutant TP63–DNA complexes to assess the alteration effects.

#### 3.6.1. RMSD Analysis

While G349 and E349 showed similar RMSD values for DNA–peptide structure, H319 showed increased RMSD values compared to the R319–DNA complex and no significant change in RMSD for native 347 and mutant 347 complexes. Increased RMSD values in R319–DNA were observed and reached an apex of 23.6 Å at 33.2 ns. Similarly, an increased RMSD value was observed between F347–DNA (24Å) and C347–DNA (20Å), and the average value was decreased for the mutant structure. The average RMSD values for G349–DNA, R319–DNA, and C347–DNA were 12.06 Å, 12 Å, and 9.82 Å respectively, whereas the average RMSD values for E349–DNA, H319–DNA, and F347–DNA were 11.03 Å, 13.09 Å, and 8.50 Å respectively (Figure 8A1, Appendix A). Simulations from 150–250 ns showed fewer fluctuations for these three mutants and wild-type structures (Figure 8A2). These wild-type–mutant pairs indicate that the RMSD was significantly higher for mutant structures than wild-type ones.

#### 3.6.2. Rg Analysis

Furthermore, Rg analysis was performed to determine the firmness and rigidity of the wild-type of DNA and mutant protein–DNA. Rg analysis showed a significant deviation in mutant H319–DNA structure, and increased Rg values were observed compared to those of the wild-type complex (Figure 8B1). The average Rg values for mutant and wild-type were 24.61 Å and 19.22 Å, respectively. The mutant 349 and wild-type 349 complexes showed a similar pattern of Rg values (i.e., average 20.93 Å and 20.99 Å, respectively) (Appendix A). The increasing trend in Rg values was also observed in the F347–DNA structure compared to wild-type, and the average Rg value of mutant 347–DNA complex (15.84 Å) was higher than that of the 347–DNA structure (13.50 Å). Simulations from 150–250 ns presented almost stable dynamics for these three mutant and wild-type structures (Figure 8B2). A closer view of the simulations is presented here for individual mutant–wild-type complexes, and it can be observed that the Rg values were higher for mutant structures compared to those of the wild-type ones.

#### 3.6.3. SASA Analysis

SASA analysis showed that mutant H319 provided increased SASA values throughout the whole analysis compared to those of the wild-type R319 complex; the average values for these two structures were 5450 Å^2^ and 5295 Å^2^, respectively. However, average SASA values for wild-type and mutant 349 were 5426 Å^2^ and 5355 Å^2^, respectively (Figure 8C1) (Appendix A). Higher SASA values represent the expansion of the structure. Hence, it can be assumed that wild-type and mutant E349, wild-type, and mutant 319 formed unstable structures with DNA. In the case of native and mutant 347–DNA complexes, a significant deviation in SASA values was observed for mutant structures. However, the average SASA value was increased to a small extent in mutant complex compared to that of wild-type. In addition, in cases of maximum SASA values, the deviation was noticed between mutant and wild-type complexes. The later simulations (150–250 ns) clearly showed that the SASA values were higher for the mutant structures compared to those for the wild-type ones (Figure 8C2). In all three analyses, the later simulations represent the approximate equilibrium state of the simulations.

The outcome of the simulation demonstrates that these three mutant DNA–TP63 complexes can be unstable after a certain time and proves the damaging effect of nsSNPs. In addition, nonbonding interaction analysis (gray, green, red/orange/purple-colored dashed lines) was performed for all the wild-type and mutant structures (Appendix A). Wild-type 319 showed several types of interactions: hydrogen bond (green color), pi-alkyl, or hydrophobic (gray, purple, orange/red), whereas mutant H319 showed fewer interactions by losing H-bonds. Mutant 349 showed fewer interactions with DNA bases compared to wild-type. The mutant 347–DNA structure also showed a smaller number of nonbonding interactions compared to native structures

### 3.7. Analysis of Non-Coding SNPs

The gene variants of TP63 with transcript ID ENST00000264731.8 were retrieved from the Ensemble database (Ensemble genome browser). The SNP source was the NCBI dbSNP database. The total number of noncoding SNPs (intron, 5′ UTR, 3′ UTR variants) was 836, with a global MAF range of 0.05–0.5.

#### 3.7.1. RegulomeDB Analysis

A total of 836 SNPs were selected for RegulomeDB analysis. The filtering process led to 21 SNPs based on the proposed ranking criteria described on the RegulomeDB site (Appendix A). Out of 23 SNPs, 20 were intron variants, one was 5′ UTR, and the remaining two were 3′ UTR variants.

#### 3.7.2. Finding eQTLs Using GTEX Analysis 

The filtered 23 SNPs from Regulome DB analysis were further analyzed with the GTEX portal to identify the single tissue eQTLs. Among 23 noncoding SNPs, five SNPs (rs4488809, rs6774934, rs6794898, rs79155799, and rs4687090) were obtained for seven eQTLs (Table 5). In addition, tissue-specific eQTLs were observed for other genes, as well. Violin plots of single-tissue eQTLs are presented in (Appendix A).

#### 3.7.3. PolymiRTS Analysis

A total of 836 SNPs were analyzed with the PolymiRTS tool. Only 44 SNPs with miRbase IDs and affecting miRNA sites were identified primarily. Those SNPs had ancestral alleles, changed alleles, and conservation scores that reflected the occurrence of miRNA sites in other vertebrates with a significant context + score change. These SNPs were classified into four functional groups: (1) D describes the disruption of a conserved miRNA site; (2) N describes the disruption of a non-conserved miRNA site; (3) C denotes the creation of a new miRNA site; and (4) O denotes without determination of an ancestral allele (Appendix A). The results show that all of the miRNA target sites for miRNA predicted to be disrupted by SNPs in TP63 were obtained from CLASH experimental data (N). In the next step, 44 SNPs were filtered considering the functional classes of C and D and conservation scores of 19–20. Finally, only five SNPs were obtained (Appendix A). During the PolymIRTS analysis, the study obtained a list of SNPs associated with disease and human tissues published in other studies (Appendix A). We matched the five miRNA binding site prediction SNPs with SNPs of previous studies, and no matches were found. Thus, these five SNPs are novel and future research can be initiated.

## 4. Discussion

TP63 is a tumor protein and acts as a transcription activator suppressor. nsSNPs of the *TP63* gene were found to be associated with different types of disorders along with cancers [56]. In recent studies, different diseases or syndromes associated with coding SNPs or nsSNP indifferent isoforms of the TP63 protein have been identified and validated [57,58,59,60,61]. The present trend favors applying a range of in silico tools rather than a single tool to classify variants as damaging or neutral [62]. We performed a comprehensive analysis of the SNPs located in the *TP63* gene using various in silico tools and approaches to obtain higher precision in the predictions of the functional impacts of the nsSNPs on the TP63 protein. Most of the in silico algorithms work based on either sequence or structural features. For example, Polyphen 2.0 uses both 3D structure and sequence to give accurate predictions [63], while SIFT and PROVEAN can work effectively based on alignment data. Additionally, PolyPhen and SNPs&GO reinforced the prediction results of the current study by incorporating structural data. Although prior studies have utilized tools such as SIFT, PolyPhen2, and CADD [64,65] as superior methods, we employed some additional tools to comprehensively characterize the missense and non-coding SNPs.

In this study, three nsSNPs (R319H, G349E, and C347F) were predicted as significantly deleterious or detrimental based on their respective prediction scores. Conserved residues are generally used in proteins to regulate the biological system, such as in folding or protein stability. Enzymatic sites in proteins contain functional amino acids involving insignificant interactions with other proteins or molecules. These residues are more conserved than the rest of the protein. From ConSurf analysis, it was observed that R319, G439, and C347 are highly conserved regions in the protein with a conservation score of 9 (Figure 4). Moreover, R319 is buried and structural, G349 is functional and exposed, and C347 is buried. Maximum conserved regions are found in the DNA binding domain, which is functionally important for DNA binding of TP63 to perform as TF. In the DNA binding domain, R319H (histidine replaces arginine) presents a small-size molecule rather than wild-type, and a positive charge was lost. Large residues are placed in the protein for both G349E and C347F (glutamate and phenylalanine replace glycine and cytosine, respectively). Moreover, a negative charge is created for G349E and in the case of C347F, the charge remains neutral. Due to the negative charge, E349 can interact with other molecules or groups, or it can result in a rejection between the mutant and neighboring residues that disrupt natural processes. This outcome was observed by HOPE. The difference in mass and charge affects the dynamics of spatiotemporal protein–protein interactions [66,67]. These three SNPs distort the contact with residues in the surroundings and hamper the usual biological activities. In addition, buried and structural residue (arginine) is more hydrophobic than histidine residue and causes loss of hydrophobic interactions in the core of the protein. Due to these altered properties of the altered residue, (H) presents a significant alteration in 3D structure of TP63, specifically, with possible loss of interactions such as H-bond interactions together with other nonbonding interactions. The altered residue is in a binding domain that is important for other molecules to bind, and it is in interaction with residues in another binding domain, activity domain, and regulatory domain. The mutation may disrupt all of these interactions among domains, affecting the protein’s function, activity, and regulation. Hence, these nsSNPs can cause loss of thermodynamic stability, aberrant folding of TP63, and aggregation with other proteins.

We executed molecular docking of the TP63 protein and DNA complex to substantiate the outcomes obtained with all the in silico tools. In a few studies, researchers showed a reduced binding affinity score for mutant protein compared to the wild-type [24,46]. The notable decrease in binding affinity for R319H and G349E was found due to H-bond loss (7/8 to 3/4) in TP63 peptide–DNA docked complexes. Therefore, the sustainability of the TP63–DNA complex is directly affected by hydrogen bond loss and binding affinity reductions that lower the DNA–protein binding probabilities. From our results, it can be predicted that nsSNPs create deleterious changes in TP63 structure and function and may potentially cause diseases such as cancers. Additionally, nsSNPs with a high binding affinity score for the mutant TP63–DNA complex represent elevated stability compared to the native complex and can be utilized to obtain efficient drugs to treat cancers. nsSNPs cause changes in protein structure. The dynamic features of proteins can be explained by MD simulations displaying additional vital features (residue plasticity and secondary structural components) that contribute to protein stability [68]. MD simulations show the protein’s real motion and structural destruction due to SNPs.

We executed MD simulations for three nsSNPs (R319H, G349E, C347). Because these three SNPs showed significantly low BA scores (Table 4) and were evaluated as pathogenic by all in silico tools, MD simulation results showed that the mutant–DNA complexes were less stable than the native protein–DNA complexes under physiologic conditions. We performed MD simulations of two buried residues (R319/H319, C347/F347). RMSD, Rg, and SASA analysis demonstrated a substantial difference in mutant structures compared to native structures. It is not possible to analyze each residual fluctuation of a nucleotide. Each residual fluctuation of the protein has been obtained from simulations but did not provide structural insight due to the lack of analysis (inclusive of DNA) of residue fluctuations of the whole complex. Hence, we did not consider plotting RMSF analysis to observe the residue fluctuations due to DNA–protein simulation. Specifically, the high average RMSD value for the R319H mutant–DNA complex was due to loss of hydrogen bond interactions. Increased RMSD values for G349E and C347F at ~0–35 ns explain the split or damage to the attached complex. The stability of the protein decreases with higher RMSD values; the low average RMSD value for the native TP63–DNA complex demonstrated the intactness of the complex, suggesting the stability of the complex (Figure 8A1). Due to the negative charge, E349 can interact with other molecules or groups. It is possible that due to increased nonbonding interactions such as electrostatic or hydrophobic interactions, RMSD was decreased. This increased stability can hamper the activity of TP63 as a TF. This is because TP63 may remain bound to DNA even if it is not required, leading to continuous transcription and abnormal production of TP63 and possibly diseases.

Simulations from 150–250 ns clearly showed the deviation in terms of RMSD, Rg, and SASA values of mutant structures compared to those of native TP63–DNA complexes. These analyses determined that the replacement of cysteine, arginine, and glycine with phenylalanine, histidine, and glutamate at respective positions in the TP63 protein significantly altered its structure from the native shape and configuration. As a result, the DNA binding affinity of TP63 protein potentially can be critically affected. Therefore, the normal functionality of the TP63–DNA complex may not be accomplished. Variations in the Rg value of mutant TP63–DNA complexes compared to native complexes may cause the TP63–DNA complex to be less compact than wild-type by enhancing the flexibility of the compound. This outcome can affect the binding of TP63 protein to DNA (Figure 8B1,B2).

The SASA findings (Figure 8C1,C2) showed that the average SASA value of mutant protein–DNA complexes always persisted to a significantly higher degree than that of the native structure in R319H and C347F mutations. The solvent-accessible surface area of the mutant–DNA complex was more expanded than that of the native TP63–DNA complex resulting in a less stable structure than the native one. This effect yields a lower probability of binding TP63 to DNA. Furthermore, the G349E mutant showed a low SASA value, indicating a less stable structure of the DNA–protein complex due to less accessibility to the DNA. However, the structural insight from mutant E349 showed that near the end of the simulation, the protein grossly engaged with DNA and showed more interactions with close packings, such as aggregations. Therefore, this was the reason for the decrease in SASA for mutant E349 near the end. The differences in the RMSD, Rg, and SASA values for native and mutant protein were further corroborated by nonbonding interactions. Hence, the evaluated nsSNPs (C347F, R319H, and G349E) have a significantly harmful effect on the stability of TP63–DNA complexes. This effect may obstruct the binding of TP63 to DNA in response to normal physiological activity, increasing the probability of causing cancers. 

Prior studies on mutant protein–ligand complexes have produced similar results with the analysis of RMSD, H-bonds, and SASA [69,70,71,72]. It was also observed that the fluctuation was high for the 0–125 ns simulation compared to the 150–250 ns simulation for these three analyses, specifically for RMSD (Figure 8A2,B2,C2). In general, the fluctuation in values of RMSD, Rg, and SASA is very common for the MD simulations, and that occurs because of structural changes over time, sometimes more drastically. The average or total values for RMSD, SASA, etc. will differ because each complex has different residues that are responsible for forming different interactions. Additionally, the simulation was approaching equilibrium, which may reach around 500 ns. However, there is a limited facility to simulate for longer than 250 ns.

The deleterious effects of R319H, G349E, and C347F are evident from the in silico analyses, molecular docking, and MD simulation. More specifically, the primary conformation of the pathogenicity of R319H was observed due to reduced DNA binding, and the outcome was obtained through molecular docking. Subsequently, RMSD, Rg, and SASA values were increased significantly compared to those of the wild-type complexes. Hence, MD simulations confirmed all the previous predictions for R319H. The outcome was corroborated by another study where research showed the association of R319H with ectrodactyly, ectodermal dysplasia, and cleft lip/palate (EEC) syndrome, and non-syndromic split hand-split foot malformation (SHFM) [73]. Hence, our study substantiated that the R319H mutant has a deleterious effect on the function and structure of TP63 protein.

PolymiRTS analysis showed five non-coding SNPs that can alter the miRNA binding sites, hampering the regulation of the TP63 protein. miRNA binds to TP63 mRNA to regulate the production of TP63 protein by inducing translation inhibition through mRNA degradation. The 3′ UTR SNPs in the *TP63* gene generate or interrupt target sites in mRNA, changing miRNA–mRNA interaction and perhaps resulting in aberrant TP63 suppression [74]. The GTEx portal allows researchers to characterize the variance in gene expression levels caused by variations (SNPs) using large sample size and a variety of human tissues [75]. QTLs play a significant role in disease phenotypes or gene expression differences by influencing the splicing process via sQTLs or the expression levels of TP63 via eQTLs [76]. Our study with the (GTEx) portal analysis showed single tissue eQTL SNPs in the *TP63* gene (Table 5) with various tissue types. The violin plots of eQTLs show normalized expression of the *TP63* gene with a specific SNP. Appendix A demonstrates the SNPs in the specific tissues and indicates the alteration in the regulation of the *TP63* gene affecting the regulation of protein products that cause diseases. Therefore, noncoding SNPs affect the normal regulation of *TP63* gene expression and protein production. These effects can result in tumor progression.

## 5. Conclusions

In this study, we initially harnessed a combination of in silico tools to collect a total of 17 nsSNPs that can alter the structure and function of the TP63 protein. Among them, 11 previously unreported nsSNPs, which were predicted to be pathogenic in our study, are crucial due to their impact on the structure and function of TP63. Nine DNA binding domain nsSNPs were finally predicted to be deleterious by all in silico tools. Molecular docking analysis of those SNPs indicated the deleterious effects by measuring the binding affinity with DNA. MD simulations revealed that R319H, G349E, and C347F are likely to make the TP63 protein structure less stable and may hinder binding with the DNA that possesses a significant effect on the function of the protein. Specifically, R319H was found to be a significantly deleterious nsSNP of TP63 protein compared to the other two i.e., G349E and C347F. The noncoding SNP analysis revealed miRNA binding site disturbance and dysregulation of the expression of the *TP63* gene in different tissue types. To confirm the association of three nsSNPs, five non-coding PolymiRTS analyzed SNPs, and five non-coding GTEx analyzed SNPs with different diseases or cancers, experimental analysis needs to be carried out in the future, i.e., performing functional analysis in cell lines. This study will serve as a benchmark for further validation of the association of TP63 SNPs with cancers.

## Figures and Tables

**Figure 1 biomolecules-11-01733-f001:**
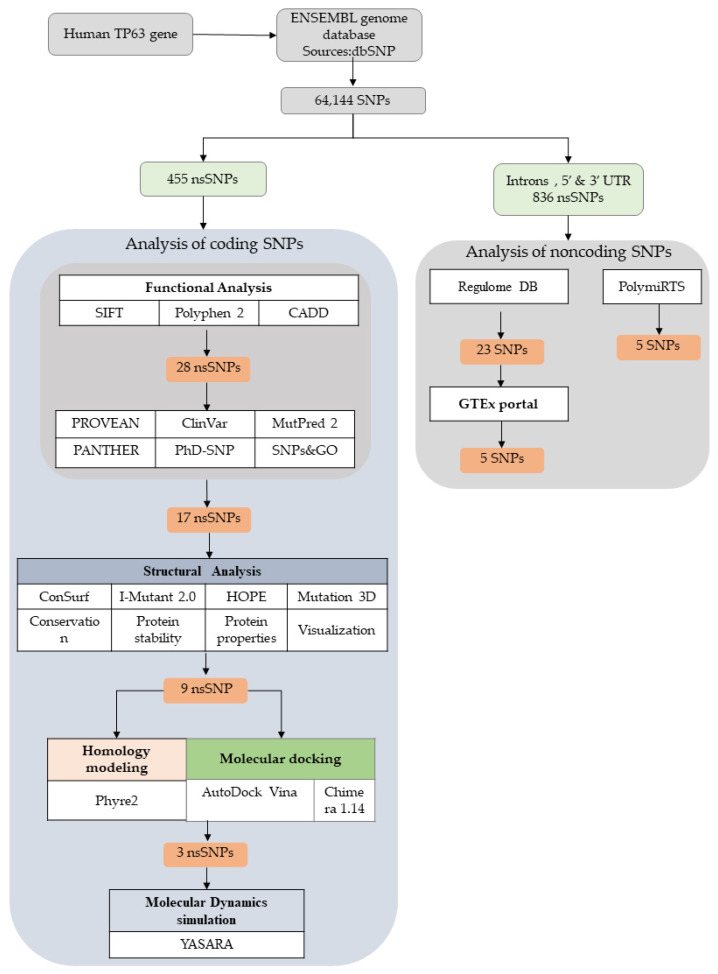
Schematic representation of the pipeline for in silico analysis of SNPs in the TP63 protein using different computational tools and algorithms.

**Figure 2 biomolecules-11-01733-f002:**
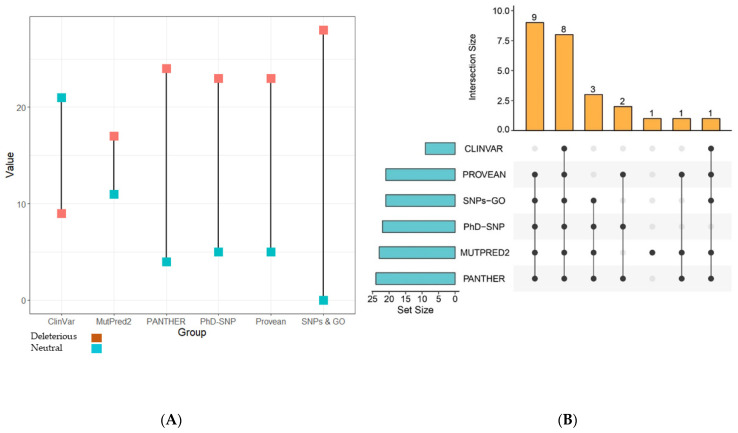
(**A**) Number of nsSNPs predicted through different Insilco tools that are described as deleterious or neutral in two different colors. (**B**) Upset plot describing a different number of deleterious nsSNPs predicted by the tools. Eight nsSNPs were predicted as deleterious by six Insilco tools. Here, connecting dots represent a combination of the tools that predicted deleterious SNPs.

**Figure 3 biomolecules-11-01733-f003:**
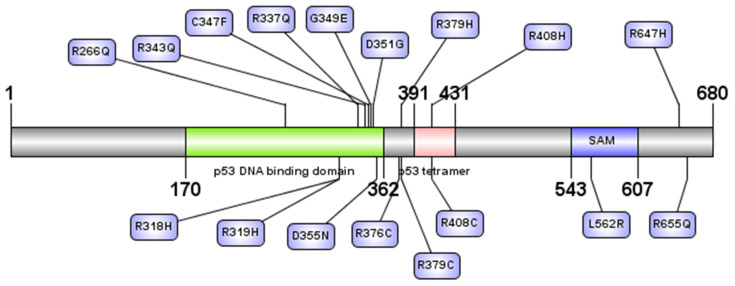
Seventeen nsSNPs that were predicted as deleterious are shown in the three principal domains of TP63 protein.

**Figure 4 biomolecules-11-01733-f004:**
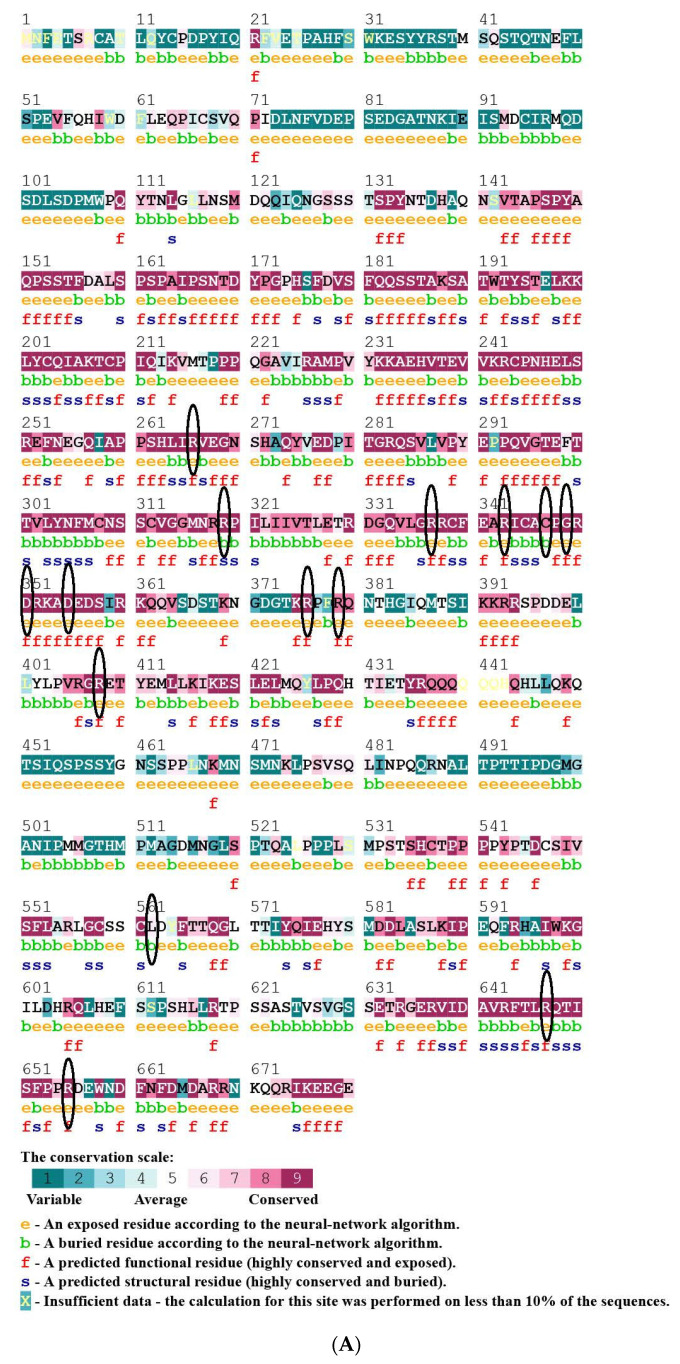
(**A**) Conservation analysis of nsSNPs using ConSurf. (**B**) Distribution of high-risk nsSNPs of TP63.

**Figure 5 biomolecules-11-01733-f005:**
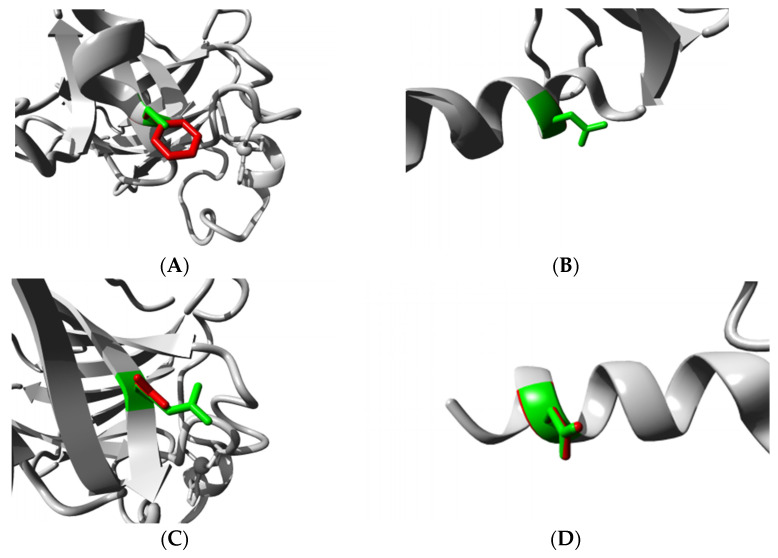
(**A**–**F**) The nine nsSNPs projected as harmful through all in silico tools. Image of the modified structure of TP63 protein was accomplished using Project HOPE. The native residue is indicated in green, and the altered residue for each SNP is depicted in red. Here, the side chain of the residue is displayed in color, and the whole protein is pictured in grey color. Notably, (**A**) C347F and (**E**) G349E show that modified residue which is in red is much bigger than native residue presented in green. In Figure 5B,C,E–I, the mutated residues are smaller than wild-type residue. In the case of (**B**) D351G, mutated G is not shown because of no side chain, and only wild-type residue is displayed in green. All of these structural changes retain detrimental impacts on the shape, conformation, and function of TP63. Note: (**A**) C347F, (**B**) D351G, (**C**) R343Q, (**D**) D355N, (**E**) G349E, (**F**) R266Q, (**G**) R319H, (**H**) R337Q, and (**I**) R318H. Figure panels (**A**–**F**) were downloaded from HOPE [42].

**Figure 6 biomolecules-11-01733-f006:**
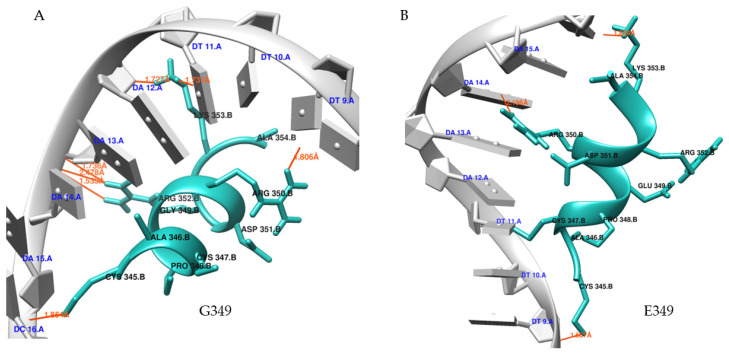
Ligand binding effect of nsSNPs through molecular docking (**A**–**F**). DNA and native/ mutant TP63 peptide complexes are shown, where DNA is represented by gray and TP63 peptide is denoted by light sea green with red-colored H-bonds. (**A**) Wild-type or native protein (G349) binding with DNA reveals that R352 has three H-bonds with DNA backbone. L353 formed one H-bond each with A12 and A11 of DNA. In addition, C345 and R350 formed one H-bond with DNA backbone and A9 of DNA, respectively. (**B**) Mutated peptide E349 formed only four hydrogen bonds. Due to SNP, E349 lost four hydrogen bonds, accompanied by loss of interaction with DNA. (**C**) Native R319 TP63 peptide generated a total of eight H-bonds. R318 formed two H-bonds each with A10 and A12 of the DNA. Additionally, R319 formed two bonds with A10 and A11 in DNA. (**D**) Mutant H319 formed only four H-bonds, and it is observed that R318 formed two hydrogen bonds with DNA sugar (A13) and H319 formed only one H-bond. This finding correlates with binding affinity reduction in the docking results, as there is a significant number of hydrogen bond loss due to SNP. (**E**) Wild-type or native protein C347 binding with DNA reveals that C345 in peptide has two H-bonds with DNA backbone, A346; I344 formed one H-bond each with DNA backbone, too. R343 and R352 also generated one H-bond each. (**F**) Mutant F347 in TP63 formed a total of six H-bonds. Most of those bonds are different than native residue, and it is observed that R350 formed two H-bonds, whereas it did not form any bond in wild-type. P348 also created bonds with DNA. New bonds are formed but original bonds are lost, which makes the structure less stable and correlates with reduced binding affinity in the docking results. Figures were created in USCF Chimera 1.14.

**Figure 7 biomolecules-11-01733-f007:**
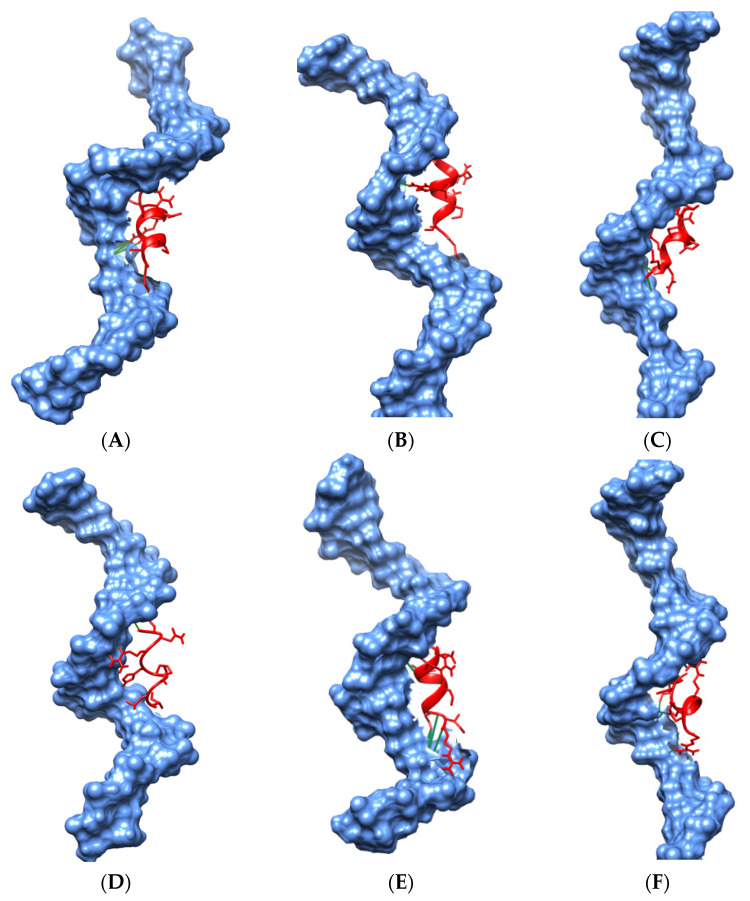
The binding site of DNA molecule with surface representation in molecular docking complexes of TP63–DNA, showing both wild-type and mutant proteins. (**A**–**F**) represent the following: (**A**) G349, (**B**) E349, (**C**) R319, (**D**) H319, (**E**) C347, and (**F**) F347 complexed with DNA. Here, blue color indicates DNA and red color denotes TP63 protein.

**Figure 8 biomolecules-11-01733-f008:**
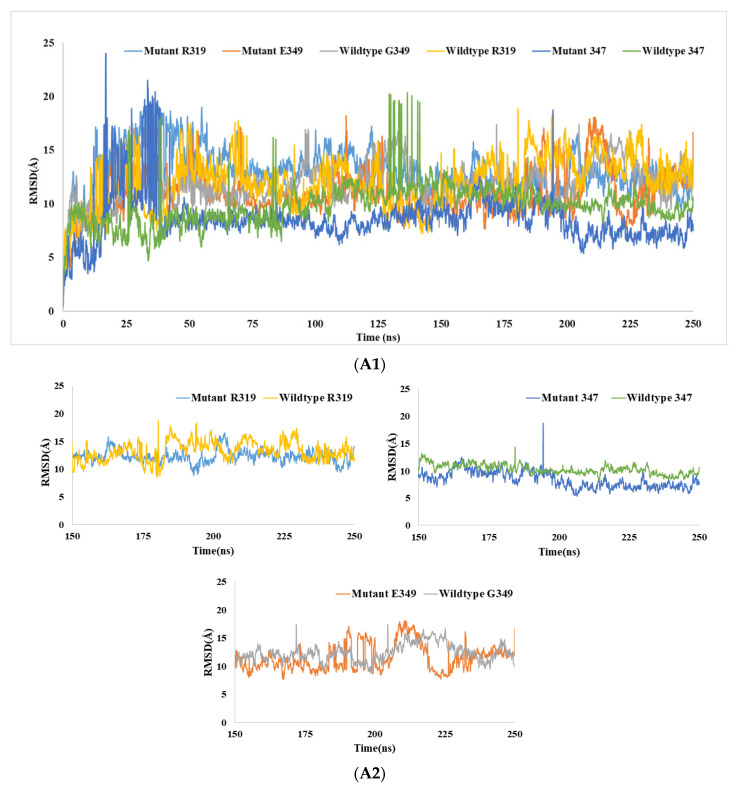
Analysis of RMSD, Rg, and SASA of wild-type and mutant TP63–DNA complexes at 250 ns. (**A1**) RMSD values of the backbone of DNA-native and mutant structures. (**A2**) RMSD values of respective wild-type–mutant pairs for 150–250ns simulations. (**B1**) Rg values of the protein–DNA backbone over the entire simulation, (**B2**) The simulations from150–250ns for respective mutant and native structures with DNA separately. (**C1**) SASA values for native and mutant TP63 with DNA, and (**C2**) the values of SASA for 150–250ns simulations for the individual pair. The symbol coding scheme is as follows: wild-type R319 (yellow color), mutant H319 (light blue color), E349 (red color), wild-type G349 (gray color), wild-type 347 (green color), and mutant 347 (blue color).

**Table 1 biomolecules-11-01733-t001:** Prediction of 17 SNPs with PROVEAN, ClinVar, PhD-SNP, PANTHER, SNPs&GO.

SNP	rsID (dbSNP)	Domain Function	Methods										
			Mutpred2		PROVEAN		ClinVar	PhD-SNP		PANTHER		SNPs&GO	
			Mdscore	Mutpred2 Impact	Score	Impact	Result	Prediction	Probability	Prediction	Probability	Prediction	Probability
R408C	rs1282887680	Oligomerization	0.662		−7.064	Deleterious	not found	Disease	0.771	Disease	0.927	Disease	0.996
R408H	rs751698974	Oligomerization	0.508		−4.461	Deleterious	not found	Disease	0.767	Disease	0.83	Disease	0.996
R376C	rs757536818	Interaction with HIPK21	0.355		−3.65	Deleterious	not found	Disease	0.716	Neutral	0.477	Disease	0.996
C347F	rs1064793282	DNA binding domain	0.932	Gain of Strand (Pr = 0.27 | P = 0.03)	−10.073	Deleterious	Pathogenic	Disease	0.91	Disease	0.975	Disease	0.997
D351G	rs121908844	DNA binding domain	0.863		−6.41	Deleterious	Pathogenic	Disease	0.836	Disease	0.916	Disease	0.997
D355N	rs1553857889	DNA binding domain/Interaction with HIPK21	0.706		−3.512	Deleterious	Pathogenic	Neutral	0.337	Disease	0.675	Disease	0.988
G349E	rs866267914	DNA binding domain	0.852	Gain-Intrinsic disorder P = 0.04 Loss of Strand P = 0.02	−7.342	Deleterious	Pathogenic	Disease	0.661	Disease	0.955	Disease	0.987
R266Q	rs121908849	DNA binding domain	0.807	Loss of Strand P = 0.02 Altered Stability P = 0.01	−3.612	Deleterious	Pathogenic	Disease	0.806	Disease	0.924	Disease	0.994
R318H	rs121908840	DNA binding domain	0.725	Loss-ADP-ribosylation at R318 P = 0.03	−4.645	Deleterious	Pathogenic	Disease	0.885	Disease	0.959	Disease	0.995
R319H	rs886039442	DNA binding domain	0.742		−4.627	Deleterious	Pathogenic	Disease	0.865	Disease	0.948	Disease	0.995
R337Q	rs113993967	DNA binding domain	0.861	Gain-Strand P = 0.02 Gain-ADP-ribosylation at R338 P = 0.05 Gain-Pyrrolidone carboxylic acid at R337 P = 0.02	−3.618	Deleterious	Pathogenic	Disease	0.8	Disease	0.901	Disease	0.995
R343Q	rs121908841	DNA binding domain	0.801	Gain of Strand P = 0.03; Altered Stability P = 0.02	−3.663	Deleterious	Pathogenic	Disease	0.855	Disease	0.902	Disease	0.996
R379C	rs761885185	Interaction with HIPK21	0.515	Loss- Intrinsic disorder P = 0.02); Loss-Phosphorylation at T382 P = 0.01; Loss-Acetylation at K375 | P = 0.01; Altered Disordered interface P = 0.04	−2.648	Deleterious	Uncertain significance	Neutral	0.424	Disease	0.602	Disease	0.995
R379H	rs765502786	Interaction with HIPK21	0.312		−1.476	Neutral	Uncertain significance	Neutral	0.227	Neutral	0.413	Disease	0.989
L562R	rs774221257	SAM	0.896	Altered Transmembrane protein P = 9.7 × 10^−5^ Altered Ordered interface P = 0.02. Altered Stability P = 0.03. Loss-Sulfation at Y564 P = 0.03	−2.328	Neutral	not found	Disease	0.787	Disease	0.517	Disease	0.998
R647H	rs774550896	Transactivation inhibition	0.834	Altered Metal binding P = 2.9 × 10^−3^. Altered DNA binding P = 1.2 × 10^−3^; Altered Disordered interface P = 0.04). Loss-Proteolytic cleavage at R643 P = 0.02. Altered Transmembrane protein P = 0.03. Altered Stability P = 0.04	−2.062	Neutral	not found	Disease	0.761	Disease	0.743	Disease	0.997
R655Q	rs764601563	Transactivation inhibition	0.656	Altered Disordered interface P = 0.04. Altered Metal binding P = 0.03. Altered DNA binding P = 0.03. Altered Transmembrane protein P = 0.05	−1.246	Neutral	not found	Disease	0.755	Disease	0.591	Disease	0.996

**Table 2 biomolecules-11-01733-t002:** Effect of nsSNPs on protein stability using I-Mutant 2.0.

SNP	DDG/∇∇G	Stability
R408C	−1.01	Decreased
R408H	−1.38	
C347F	−0.48	
D351G	−1.64	
D355N	−1.49	
G349E	−1.48	
R266Q	−1.03	
R318H	−1.3	
R319H	−1.38	
R337Q	−0.91	
R343Q	−0.99	
R379C	−0.41	
R379C	−0.14	
L562R	−1.84	
R647H	−1.97	
R655Q	−1.96	

**Table 3 biomolecules-11-01733-t003:** Prediction of structural and functional consequences of nsSNPs in TP63 through Project HOPE.

SNPs	Size & Charge		Characteristics & Features
	Wild-Type	Mutant	
R376C	Large & (+ve)	Small & (0)	Hydrophobicity: High. Effect: High. Protean folding: Affected. Loss of interaction: High & distributed;
R4(0)8C	Large & (+ve)	Small & (0)	
R4(0)8H	Large & (+ve)	Small & (0)	
C347F	Small & (0)	Large & (0)	
D351G	Large & (−ve)	Small & (0)	
D355N	Large & (−ve)	Small & (0)	
G349E	Small & (0)	Large & (−ve)	
R266Q	Large & (+ve)	Small & (0)	
R318H	Large & (+ve)	Small & (0)	
R319H	Large & (+ve)	Small & (0)	
R337Q	Large & (+ve)	Small & (0)	
R343Q	Large & (+ve)	Small & (0)	
R379C	Large & (+ve)	Small & (0)	
R379C	Large & (+ve)	Small & (0)	
L562R	Small & +(0)	Large & (+ve)	
R647H	Large & (+ve)	Small & (+ve)	
R655Q	Large & (+ve)	Small & (+ve)	

(+ve): Positive, (−ve): Negative, (0): Neutral.

**Table 4 biomolecules-11-01733-t004:** Binding affinity of nine nsSNPs using molecular docking.

Wild-Type		Mutant		
Residue	Binding Affinity	Residue	Binding Affinity	Binding Affinity Change
R266	−6.2	Q266	−5.9	Decrease
R318	−5.3	H318	−6.2	Increase
R319	−5.8	H319	−5.4	Decrease
R337	−5.5	Q337	−5.8	Increase
R343	−5.8	Q343	−5.8	Neutral
C347	−6	F347	−5.6	Decrease
G349	−6.4	E349	−5.8	Decrease
D351	−5.6	G351	−5.4	Decrease
D355	5	N355	−5.8	Increase

**Table 5 biomolecules-11-01733-t005:** eQTLs prediction of non-coding SNPs in GTEx portal.

Gene Symbol	Variant ID	SNP ID (Non-Coding)	*p*-Value	NES	Single Tissue eQTL
TP63	chr3_189638472_T_C_b38	rs4488809	6.5 × 10^−7^	0.23	Lung
TP63	chr3_189638472_T_C_b38	rs4488809	1.7 × 10^−5^	0.13	Nerve-Tibial
TP63	chr3_189664468_A_G_b38	rs6774934	5.5 × 10^−5^	−0.34	Heart-Left Ventricle
TP63	chr3_189664468_A_G_b38	rs6774934	6.5 × 10^−5^	−0.20	Nerve-Tibial
TP63	chr3_189672911_G_C_b38	rs6794898	1.3 × 10^−5^	0.20	Lung
TP63	chr3_189710792_T_G_b38	rs79155799	6.5 × 10^−7^	−0.17	Nerve-Tibial
TP63	chr3_189721190_A_G_b38	rs4687090	1.7 × 10^−5^	−0.19	Nerve-Tibial

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
