# Peer review of "Comprehensive Characterization of the Coding and Non-Coding Single Nucleotide Polymorphisms in the Tumor Protein p63 (TP63) Gene Using In Silico Tools"

_biomolecules, 2021, doi:10.3390/biom11111733_

Round 1
Reviewer 1 Report
This paper deals with the effects that Single Nucleotide Polymorfisms (SNPs) have on the interaction between DNA and TP63. It is well known that defects in the structure of this protein lead to varies form of cancer and, for such reason, predictions on possible, SNPs-induced changes in the TP63 binding capability to DNA are relevant for human health.
I think that results obtained match the main goals declared by the authors in the introduction of their manuscript, even if it is a real pity that the whole paper is purely theoretical. Checking at least the structural stability of a few samples through simple temperature- (or pH-, or salts-) dependent measurements would have considerably improved the robustness of their predictions.
Nonetheless, I think that the large number of information obtained on the potential harmfulness of the SNPs studied justify the publication on Biomolecules.
I have only minor points/questions for the author:
Despite the whole tetrameric TP63 structure is not yet available, I guess that it could be interesting to know whether (and to which extent, if any) the SNPs which produce the largest effects on the DNA-TP63 interaction might alter the 3D structural features of the dimeric TP63 DNA binding domain (for instance at the level of the subunit interface).
Which is the reason to report so many decimals in the SASA, RMSD and Rg values?
For instance, -5450.324 Å2 is nonsense: it should be approximated to -5450 Å2…..Rg=24.611 Å should be 24.6 Å and so on….
Please select only one type of amino acid code (one letter or three letter, not both)
Figure 4 should be printed with a better resolution
Please note that there are several typos that should be removed (for instance in Figure 4; or the term “Wild type” that sometimes is written in capital letters, sometimes it is not….)
Author Response
Dear Reviewer,
Greetings!
First, thank you for reviewing our manuscript and suggestions to improve it. We thoroughly reviewed your comments and addressed all of them in the revised manuscript.
Please find the attached document indicating and explaining how we addressed your feedback in the manuscript. We are open to any suggestions going forward.
Thank you for your time.
On behalf of the authors,
Shamima Akter

Reviewer 2 Report
The authors provide a comprehensive in silico analysis of TP63, a gene associated with several oncological diseases. They use a wide array of methods, further confirming and refining their conclusions as the evidence of disease causation and deleteriousness increases for each SNP. I believe the work here presented is good but can become better (for clarity, I will be commenting mostly on the field where I can offer greater criticism - computational structural biology). I ask the authors to address the following commentaries:
Major comments:
- The explanations for the computational methods provided in 2.2 can be too vague and are hard to relate to the actual methods. It would be more useful for the methods to also include a detailed explanation of the flowchart in Figure 1
- Tables 2, 3 are often repetitive and have quite a bit of redundant information... I recommend the authors find a more appealing way of presenting these data
- Figure 4A needs to be in much higher resolution, right now it is barely readable and takes away from the message of the paper
- The MD analysis, while informative, could be more informative - why not analyse specific residues of interest for the buried surface area? In other words, while the average or total values for RMSD/SASA/etc. may be different, what exactly is causing this difference? And what do the authors make of the decrease in SASA for Mutant E349 near the end?
- In this work, the focus of the discussion is on the methods and not so much on offering a good oversight of the results. I believe it would be considerably easier to read the discussion if the authors concentrated on the residues for which they have good evidence of deleteriousness or association to disease since they already provide a good explanation of the results
- The figures feel very disconnected from one another... If the authors have time for this, I would appreciate it the figure presentation was made more homogeneous. Additionally, having single figures spanning multiple pages makes it quite complicated to follow
- I would appreciate it the reviewers could review the English used through this manuscript since some parts contain grammar errors and make the text complicated to analyse appropriately
Minor comments:
- There are several parts where the English, while not being poor, could be clearer. I would advise the authors to calmly read through their work and perhaps make a few alterations to create a more reasonable flow and cadence for the article. For example, when the authors write "In CADD, likely deleterious criteria have been chosen and 28 nsSNPs were predicted to be deleterious", what they surely mean is "28 SNPs were classified as likely deleterious by CADD." Such corrections would make this manuscript much easier to read
- Some parts of the text have no spaces between words or between words and numbers --- I ask the authors to please go over the manuscript and correct this accordingly
- The authors should introduce the concept of SNP more clearly and describe the role and function of TP63 in the introduction. Additionally, they should try to explain why non-coding SNPs are important
- When the authors mention that "Non-synonymous SNPs (nsSNPs) might have a greater impact on the protein structure, function, stability, and solubility" - what are they comparing this with? What do they mean by "might have"?
- Gene names have to be in italic
- The authors should explain what DeltaDeltaG is
- YASARA is wrongly spelled in Figure 1, please correct this
- In the results, when the authors describe the Consurf results, they say that "Four residues were found to be medium conserved where two residues (R318, R379) are functional and exposed, one is structural and buried C347, and the other one is buried." they should mention which one is buried. Additionally, please provide clarification about the meaning of "structural" in this context
- In the HOPE analysis section, the sentence "Size of a.a residue was shown to change to bigger/smaller and the charge was observed to lose/gain." is hard to understand, I ask the authors to please make it clearer. Additionally, hope results are not predictions, they're just the result of a simple analysis
- "Mutant" is misspelled in Figure 8A2
- I may be missing something but there is not Table 5
- The section with PolymiRTS is confusing - this a database of miRNA binding sites from other studies, so it is only natural that they will be found in other studies. When the authors say "This SNP has been previously published in a study" in this section what are they trying to add? Please clarify what aspect of this analysis entails novelty and why it is essencial to this work
- The authors alternate between "in silico" and "in-silico", please settle on one
- What do the authors mean by "nsSNPs possess observed changes in conformation for protein" in the discussion?
Author Response

(The authors gave the same response as above.)

Reviewer 3 Report
The authors used a variety of computational tools to analyze the structural and functional effects of SNPs on TP63 genes and they concluded that eight nsSNPs are highly conserved in the protein and are predicted to reduce protein stability and adversely affect TP63 protein function. They also discovered that six non-coding SNPs and GTEx portals at miRNA binding sites identified five eQTLs SNPs in individual tissues of lung, heart (LV), and cerebral hemisphere (Brain). Here comes some comments.
Major comments:
- In page 15, why just three nsSNPs were selected for MD simulations. Form Table 4, a significant binding affinity change had occurred for D355 between wild type and mutant. It needs a binding affinity change column to show the changes between wild type and mutant.
- The corrected P-value analysis is preferred in Table 6.
Minor comments:
- Some parts of the text are indented, and some parts are not. Please be consistent.
- Figure 4 is very unclear.
- Some references have DOI numbers and some do not.
Author Response

(The authors gave the same response as above.)

Round 2
Reviewer 2 Report
I was only going to suggest the authors go over the English in their manuscript one more time.
Reviewer 3 Report
No